# Reflections on contributing to health research: A qualitative interview study with research participants and patient advisors

Lisa Newington[1,2]*, Caroline M. Alexander[1,2], Pippa Kirby[2], Rhia K. Saggu[2], Mary Wells[1,3]

1 Department of Surgery and Cancer, Faculty of Medicine, Imperial College London, London, United Kingdom, 2 Therapies, Imperial College Healthcare NHS Trust, London, United Kingdom, 3 Nursing Directorate, Imperial College Healthcare NHS Trust, London, United Kingdom

* l.newington@imperial.ac.uk

## Abstract

### Objectives

The aims of this study were to explore individuals' experiences of contributing to health research and to identify the types of impact that are perceived as important by participants or patient and public advisors. Specifically, research led by NMAHPP clinicians (Nursing, Midwifery, Allied health professions, Healthcare science, Psychology and Pharmacy).

### Methods

Semi-structured one-to-one interviews were conducted with health research participants and patient or public advisors. Interviewees were recruited from five UK sites and via social media. Interview transcripts were analysed using Thematic Analysis to identify key themes and areas of disagreement.

### Results

Twenty-one interviews were completed, and four main themes were identified. The first, optimising research experiences, included personal reflections and broader recommendations to improve participant experiences. The second, connecting health research with healthcare, described research as key for the continued development of healthcare, but illustrated that communication between research teams, participants, and clinicians could be improved. The third theme explored the personal impacts of contributing to research, with interviewees recalling common positive experiences. The final theme discussed capturing research impacts. Interviewees highlighted potential priorities for different stakeholders, but emphasised that financial impacts should not be the sole factor.

### Conclusion

Individuals who were involved in NMAHPP health research recalled positive experiences and reported good relationships with their research teams. They felt that their contributions were valued. Suggested strategies to optimise the research experience focused on

**Data Availability Statement:** Supporting study documents are provided via the OSF repository https://osf.io/wurz3/. Where permission was granted, data supporting this study are available

from the ReShare Repository hosted by the UK Data Service: Newington, Lisa; Alexander, Caroline M; and Wells, Mary (2022). What does research impact mean to participants? Transcripts and coding content from 17 qualitative interviews. [Data Collection]. Colchester, Essex: UK Data Service https://reshare.ukdataservice.ac.uk/855956/. Full access to the data is subject to approval and a data sharing agreement.

**Funding:** LN was funded by an NIHR Imperial Biomedical Research Centre and Imperial Health Charity Postdoctoral Research Fellowship (RFPO2122_3). RKS was funded by an NIHR Imperial Biomedical Research Centre Pan-London Research Fellowship (ICHT-39). PK was supported by Imperial Health Charity, MW was supported by the NIHR Imperial Biomedical Research Centre and infrastructure support for this research was provided by NIHR Imperial Biomedical Research Centre. The funders had no role in study design, data collection and analysis, decision to publish, or preparation of the manuscript.

**Competing interests:** The authors have declared that no competing interests exist.

simplifying documentation, clear signposting of the research activities involved, and feedback on the research findings. Routine sharing of relevant research data with clinicians was also recommended. Personal impacts included a deeper understanding of their health condition or health more broadly, and increased confidence interacting with healthcare and other professionals. These findings will be used to inform development of a framework to capture the impact of NMAHPP research.

## Introduction

The translation of research findings into effective clinical practice is a universal healthcare challenge [1]. Strategies to increase research implementation include aligning research questions with clinical and patient priorities [2], and embedding research activity within clinical practice [3–6]. Research-active healthcare organisations have consistently been found to have better patient outcomes than their non–research-active counterparts [7–10].

There is widespread recognition that increasing research activity among NMAHPP clinicians is an important step to support the generation of relevant research and facilitate the implementation of research findings into practice [11–19]; NMAHPP describes Nursing, Midwifery, Allied health professions, Healthcare science, Psychology and Pharmacy. However, in comparison to the medical professionals, there is currently a lack of structure and access to clinical academic career pathways for the NMAHPP disciplines [20–22]. Various initiatives have been reported, such as embedded researcher models [23], practice-based research training programmes [24], interdisciplinary internship schemes [25], and organisation- and discipline-specific postdoctoral career pathways [26].

Along with increasing research activity, there is also a need to capture, demonstrate and evaluate the impact of the resulting research. We recently explored the reported impacts of research activity by NMAHPP clinicians [27, 28]. Identified themes contained several proposed benefits to patients, including increased access to evidence-based management, improved care pathways and service delivery, and changes in the local culture that promoted patient-focused care. However, these proposed categories of impact were reported by clinicians and managers, and it is not known how patients or research participants perceive the impact of such research activity. When capturing and evaluating the impact of research, it is important to consider the features that have been identified as meaningful to participants and patient or public contributors as well as those perceived as valuable to researchers, clinicians or healthcare organisations.

The UK National Institute for Health Research (NIHR) terminology refers to research 'participation' as taking part in a study, for example being recruited to a clinical trial or interview study [29]. Research 'involvement' is defined as research being carried out with or by members of the public [29]. This includes as members of a study advisory group or being involved in data collection or analysis. The NIHR 2018/19 Research Participation Experience Survey found that ~90% of participants reported a good experience of taking part in research [30]. However, the survey predominantly collected quantitative data, which did not allow in-depth exploration of the factors contributing to this response. More recently, the NIHR 2020/21 Public Involvement Feedback Survey found that ~80% of participants were satisfied or very satisfied with their experience [31]. Reasons for dissatisfaction included the difficulty of participating and developing research relationships in virtual meetings, particularly when these were infrequent [31].

Existing literature on research participation primarily focuses on the reasons why patients did or did not agree to participate in, or complete, a research study [32–35]. This is valuable in

informing optimal study designs, including processes to maximise recruitment and minimise loss to follow-up, however, to the best of our knowledge, the concept of participant-perceived research impact has not been previously explored outside feedback for individual research studies [36].

The aim of this qualitative interview study was to explore individuals' experiences of contributing to healthcare research with the objective of identifying the types of impacts that are important for participants or patient and public advisory group members. We were particularly interested in healthcare research that was led by NMAHPP clinicians as this qualitative study forms part of a larger programme of work to develop a research impact framework for clinical NMAHPPs.

## Materials and methods

### Ethics approval and consent to participate

The study was approved by the NHS Research Ethics Committee and Health Research Authority (IRAS 298078, REC 21/WA/0229). Local permissions were in place for study invitations to be emailed to eligible individuals aged ≥18 years via Patient Identification Centre agreements. In addition, permission was granted for study information to be posted on social media platforms for participating organisations. All interviewees provided written informed consent via an electronic consent form.

### Research team and reflexivity

The core research team comprised experienced mixed-methods researchers who were clinicians in physiotherapy (LN, CMA) and nursing (MW). Additional support with data coding and preliminary analyses were provided by novice qualitative researchers from dietetics (RKS) and speech and language therapy (PK). Two patient and public advisors contributed to the initial study design and supported the development of study materials (FA, JL) and one public advisor contributed to the analysis (FA). Contextually, this research was situated within a realist perspective [37]. Through this paradigm, we sought to explain the phenomenon of research impact through the individual views and experiences of our interviewees, with the interpretation that these individual experiences also contribute to a broader picture. This included the exploration of three aspects: experiential elements (describing interviewees' subjective experiences and views); interferential elements (what might be inferred from these experiences); and dispositional elements (attempting to theorise about the phenomenon) [38].The Standards for Reporting Qualitative Research (SRQR) checklist [39] was used to guide reporting (S1 File).

The core research team had previously systematically reviewed NMAHPP research impact [27] and interviewed healthcare managers and research-active NMAHPPs about their experiences of research [28]. Coding and analysis in the current study was not based on the findings from this previous work. We used an inductive coding approach with no preestablished coding ideas, and the initial coding and final discussions around themes and sub-themes were supported by individuals not involved previously (RKS, PK, FA). However, we acknowledge that our own research experiences and awareness of the existing literature will have framed our approach.

### Participants and recruitment

Five Participant Identification Centres (PICs) were recruited; members of these research teams shared the study invitation with their current and past research participants and patient and public involvement (PPI) groups. In addition, the study was advertised via social media by

**Table 1. Study eligibility criteria.**

| Inclusion criteria | Exclusion criteria |
|---|---|
| Adults (≥18 years) and able to provide informed consent. | Under 18 years or unable to provided informed consent. |
| Current or previous contributions to healthcare research within the past 3 years. This could be as a participant or as part of a patient and public involvement (PPI) group. | No contributions to healthcare research in the past 3 years. |
| Contributed to research led by NMAHPP clinicians: Nursing, Midwifery, Allied health professions, Healthcare science, Psychology or Pharmacy. | Only contributed to research led by clinical doctors, dentists or non-clinical university researchers. |
| Able and willing to participate in a single phone or video call interview in English language. | Research was led or entirely delivered outside the United Kingdom. |

five sites. Interested individuals were invited to complete an electronic expression of interest form, which included background demographic and eligibility information. These individuals were contacted by the lead researcher to confirm eligibility and arrange an interview. In the case of non-response, up to two reminders were sent. Eligibility criteria are provided in Table 1. None of the interviewees were known to the research team in either clinical or research capacities.

Recruitment continued until the research team agreed that data saturation had been achieved. This was operationalised using a blended definition: no new items identified during the interview, no new codes identified during initial coding, and agreement that sufficient data had been collected to adequately address the research aims [40]. The feasible sample size was anticipated to be between 15–25 interviews. No reimbursement was provided to interviewees.

## Study design

Semi-structured interviews were led by the first author using a pre-piloted interview guide that was developed in collaboration with two patient and public advisors (JL, FA). Questions focused on individual experiences of being a participant, including any benefits or challenges of their research contribution; ideas about what makes good healthcare research; views on the important impacts of healthcare research; and how these might be recorded. The full interview guide is provided via the Open Science Framework [41].

As this was a national study, interviews were offered remotely via phone or video call according to interviewee preference. Interviews were audio recorded and transcribed verbatim by an external transcription company (PageSix Transcription), bound by a non-disclosure agreement. Completed transcripts were checked against the audio file and anonymised to remove names, places, medical conditions and other potentially identifiable details. All interviewees were offered the opportunity to review their transcript and provide feedback.

Data were analysed using Reflexive Thematic Analysis [42, 43]. Initial inductive coding was completed by LN, RKS and PK on four transcripts. The preliminary codes were discussed and refined with input from all authors. An operational coding tree was developed and applied to the remaining transcripts by the lead author. This was not a rigid codebook, rather a method of keeping track of the coding thoughts and descriptions as the research team were all working on multiple different projects. Additional codes were added and defined through discussion with all authors. Codes were also redefined, relabelled, merged, and expanded as part of this iterative analysis process, which involved reviewing and recoding earlier interviews in light of changes to the coding content and structure. The final coding tree is available from the Open Science Framework [41]. Codes were categorised into initial themes and sub-themes, which were discussed and refined by the authors and through feedback from a patient and public

advisor (FA). During development, the themes were reviewed to ensure that they represented both consistent and differing views among interviewees. Data for research participants and patient advisory group members was not handled separately during the analysis. However, the composition of the themes and sub-themes was explored and any differences in perspective based on interviewee demographics (including type of research contribution) was discussed within the theme.

## Results

Forty-three individuals expressed an interest in the study. Of these, 11 were ineligible: nine because they had been involved solely in research led by clinical doctors, and two because their research contribution was outside the UK. A further nine did not respond to the interview invitation and two scheduled interviews but were unable to participate during the study period due to ill-health.

An estimated 574 email invitations were sent by the PIC teams, frequently as part of a study update or general newsletter, and the study was advertised on five organisation-level social media platforms. Interviews were conducted between October 2021 and April 2022; 18 by video call (Zoom), one by Zoom audio call, and two by telephone. Mean interview duration was 45 minutes, range 23–86 minutes. Eight participants opted to review their transcripts after the interview; no changes were requested.

Interviewee characteristics are provided in Table 2 and compared with those who expressed an interest and were eligible to take part, but were not interviewed due to non-response or ill-health.

Interviewees' reflections on their research contributions were illustrated as four key themes: 1) optimising research experiences, 2) connecting health research with healthcare, 3) personal impacts of contributing to research and 4) capturing research impacts. Each theme comprised several sub-themes, which were labelled using direct quotes from interviewees. Sub-themes explained components of the main theme and described areas of disagreement among interviewees, where this arose. Illustrative quotes are provided in the text and Table 3, with interviewees identified as number 1–23. Interviews 14 and 17 did not take place due to interviewee health issues. A thematic representation of the identified themes in shown in Fig 1.

### 1. Optimising research experiences

Interviewees provided their perspectives of what makes good health research. 'Good' was defined in terms of societal benefits and participant experiences, and responses were categorised into four sub-themes which included both reflections on personal experiences and broader recommendations. The first sub-theme focused on the need to address an important or meaningful research question *(1.1 Add something to what is known already)*. The second highlighted the importance of ensuring that research participants and other contributors were valued for their commitment *(1.2 Made to feel that what I think counts)*. The third focused on *Research made simple (1.3)*, which included accessible documentation, ease of attending or contributing to the various research activities, and easy access to the research team. The final sub-theme centred on reimbursement for participation *(1.4 It is offered, and I think actually that's critical)*.

**1.1 *"Add something to what is known already"* (#23).** The importance of creating new knowledge was widely suggested as a key feature for health research. This included avoiding repetition of previous research and specifically looking for innovation and novel collaborations. Interviewees also suggested that the anticipated knowledge and outcomes from each study should be clearly communicated with potential participants.

**Table 2. Characteristics of interviewees compared with non-responders.**

| | Interviewees n = 21 (%) | Non-responders or unable to participate n = 11 (%) |
|---|---|---|
| *Gender* | | |
| Male | 13 (62) | 3 (27) |
| Female | 8 (38) | 8 (73) |
| *Ethnicity* | | |
| Asian / Asian British | 1 (5) | 1 (9) |
| Black / African / Caribbean / Black British | - | 2 (18) |
| White / White British | 19 (90) | 7 (64) |
| Other | 1 (5) | 1 (9) |
| *Mean age in years [range]* | 62 [33–93] | 47 [22–78] |
| *Age category (years)* | | |
| $\leq$ 49 | 5 (24) | 7 (63) |
| 50–69 | 8 (38) | 2 (18) |
| 70–89 | 7 (33) | 2 (18) |
| $\geq$ 90 | 1 (5) | - |
| *Geographical location* | | |
| East of England | 1 (5) | - |
| London | 4 (19) | 5 (45) |
| North East and Yorkshire | 8 (38) | 2 (18) |
| North West | - | - |
| South East | 1 (5) | 2 (18) |
| South West | 5 (24) | - |
| Wales | 2 (10) | - |
| Scotland | - | 2 (18) |
| *Type of research contribution* | | |
| Participant | 9 (43) | NR |
| Patient or public involvement | 4 (19) | NR |
| Both | 8 (38) | NR |
| *Clinical discipline leading most recent research contribution** | | |
| Clinical psychology | 4 | 1 |
| Dietetics | 1 | 1 |
| Healthcare science | - | 1 |
| Occupational therapy | 1 | - |
| Pharmacy | 4 | 1 |
| Physiotherapy | 6 | 2 |
| Podiatry | 1 | - |
| Speech and language therapy | 5 | 1 |
| Unsure | 1 | 5 |

* Multiple responses allowed. NR Not recorded.

> "So, tell me what the research is, tell me why you're doing it. And tell me what you hope to get from it." #6, male, 50–69 years, participant and PPI.

> "It would be good to know a bit more about what they actually hope to do with it, in terms of improving people's outcomes or whatever it might be." #7, male, <49 years, participant.

**Table 3. Additional quotes to illustrate each theme and sub-theme.**

| Theme 1. Optimising research experiences | |
|---|---|
| 1.1 Add something to what is already known | *"I am interested in–might the project add something to what is known already."* #23, F, 50-69yrs, participant & PPI. |
| | *"I think asking people, ask people, you know, when they first get involved what they think would help."* #11, F, 50-69yrs, participant & PPI. |
| | *"I feel strongly about projects recruiting people with lived experience of whatever is under discussion."* #23, F, 50-69yrs, participant & PPI. |
| 1.2 Made to feel that whatever I think counts | *"Every research that I've been in, piece of research that I've been involved in, I've been made to feel that whatever I think counts, whether it's completely gobbledygook. . . That it's special, just as the person who follows me is special, and the person that preceded me is special, that they've contributed to this well, this pool, and it's that feeling that it might make a difference somewhere else further down the line."* #20, M, 70-89yrs, participant & PPI. |
| | *"But if there was a nice, easy patient-focused dissemination of results, so with an, I guess, upbeat way of saying, "Hey, this is really important for us to know, so we can use this for future research.""* #15, F, ≤49yrs, participant & PPI. |
| | *"I think that's really important, as I said all the way through it is that feeling of you having a real seat at the table and you're valued, you're worthwhile."* #6, M, 50-69yrs, participant & PPI. |
| 1.3 Research made simple | *"Then you say, 'And the toilets are at the end of the corridor on the left'."* #13, M, 70-89yrs, participant & PPI. |
| | *"I've not being going in routinely to the hospital since Covid so as it turned out there were a few things they needed to get done anyway so they sort of tagged on the tests."* #7, M, ≤49yrs, participant. |
| | *"We're missing the point here because we're doing this and really all we need is a little sentence, a brief description of what's going to happen and then a tick in a box and a signature."* #6, M, 50-69yrs, participant & PPI. |
| 1.4 It is offered, and I think that's actually critical | *"Where I am able to decline, I do. There are sometimes where they say because of the process we have to actually, and I get paid, and that goes to charity. . . It is offered, and I think that's actually critical. Not everybody's fortunate enough to not need it."* #21, M, 70-89yrs, PPI. |
| | *"The study I have been involved with before and this one, I have been able to fit it in and I don't need financial recompense. If it was something bigger, deeper, longer-term–if there were some kind of recompense yes sure, I would be happy to accept that! Work is up and down, isn't it!"* #1, M, 50-69yrs, participant. |
| | *If we're interested in the issue and see the sort of public value and we've got the time, then it's worth doing. [Payment] is a bonus really from our point of view. . . I think ultimately the major motivation is unlikely to be financial. It will be that you feel better about having done it."* #12, M, 60-79yrs, participant. |
| **Theme 2. Connecting health research with healthcare** | |
| 2.1 It's a good relationship | *"I have the greatest respect for these people [research-active clinicians], as you will have gathered."* #4, M, 70-89yrs, participant. |
| | *"They usually are incredibly polite, pleasant to deal with and grateful for what one contributes."* #5, M, 70-89yrs, participant. |
| | *"This really gave me an interesting look into sort of like behind the scenes and how they're sort of trying to further things, trying to get the best way of dealing with things."* #10, M, ≤49yrs, participant. |
| | *"I've learnt the hard work that goes into people trying to improve a nation's health, so I respect that also, it gives me a respect of what people are trying to do."* #19, M, 70-89yrs, PPI. |

(*Continued*)

**Table 3.** (Continued)

| | |
|---|---|
| 2.2 People at the frontline doing their own research | *"It just seemed to be a natural part of working with a very disciplined and highly communicative multi-disciplinary team. It just seemed a natural part of it. I know it was [clinical academic researcher's] instigation and her specialism, it was seamless. #1, M, 50-69yrs, participant.* |
| | *"I guess there needs to be I think time made for it otherwise you'll just perpetuate doing what's currently being done and not really acting to the fact that whatever technology, new drugs, will change dramatically. . . It is obviously difficult to manage that, but I think it should be a reasonably high kind of priority in terms of learning how to design the service for the future really." #7, M, ≤49yrs, participant.* |
| | *"Sadly, we can't clone clinicians, I mean, that would be the ideal solution, because I've often found it's the best clinicians who are aware of the limitations in their own practice who go into research to actually improve practice. So yes, it's hard." #11, M, 50-69yrs, participant & PPI.* |
| 2.3 Communication is everything | *"I feel that there is a problem for research in general to have a clearer line of communication with individuals about what they are hoping to do as well as telling them what they have done." #16, M, ≥90yrs, participant & PPI.* |
| | *"It makes us feel as if we've done something quite good, it makes us feel good that we're helping somebody and some project, and you may need us in three years' time or six years' time or something like that." #13, M, 70-89yrs, participant & PPI.* |
| **Theme 3. Personal impacts of contributing to research** | |
| 3.1 The satisfaction is that you help somebody | *"I mean I am very keen to help, I have had so much benefit from that hospital and I am really keen to do anything I can to help them in return." #4, M, 50-69yrs, participant.* |
| | *"It just felt like a good thing to do." #7, M, ≤49yrs, participant.* |
| | *"I do it because I feel I gain a lot of interest myself. . . I want to do it because I think it's worthwhile." #16, M, ≥90yrs, participant & PPI.* |
| 3.2 Socialisation | *"And the important bit, from my point of view, was meeting people, different people from different parts of the world." #13, M, 70-89yrs, participant & PPI.* |
| | *"Mixing with people that were at different stages of their recovery and had different rates of recovery and completely different sort of outcomes as well, and so that's given me more knowledge". #10, M, ≤49yrs, participant & PPI.* |
| | *"And we all had such energy and commitment, and we were a varied bunch, so we were bringing lots of ideas, but because we'd all had the relevant patient experience, we had understanding and empathy for each other." #11, F, 50-69yrs, participant & PPI.* |
| 3.3 Deepened my understanding | *"Because it was more thorough than perhaps it would have been otherwise, so it just deepened my understanding. The team are like that anyway, anything you want to know you just ask." #1, M, 50-69yrs, participant.* |
| | *"I got a nice booklet, well a pamphlet really, telling me about me, and somebody of my age, and a comparison with other people of my age, so that told me a little bit more." #20, M, 70-89yrs, participant & PPI.* |
| | *"I felt stimulated. I would say that listening to researchers and the subjects that they present to us and I find I have a much broader spectrum of understanding." #19, F, 70-89yrs, PPI.* |
| 3.4 From that I realised you have to ask questions | *"It is quite important for self-esteem, people who have felt marginalised by the process that they haven't always felt listened to by healthcare professionals or that their condition has been brushed over as less important than some other conditions. So, this thing of being heard and their viewpoint validated I think is an important outcome for a lot of the people who are involved in projects." #23, F, 50-69yrs, participant & PPI.* |
| | *"I took all of this data, I put it all together as a presentation sort of thing and said to my consultant, 'I want to mix [medications]'. And he went, not in [that format]. . . Then I went, 'I've got this data, mate. Have a look at it'. I said, 'look, my [measurement] has improved by. . . 'I think it was about 14% or something. It wasn't massive, but it was a noticeable difference, and when you draw green graphs, it was a very noticeable difference. And he was like, 'yeah, yeah, I'll sign off on that'. So, that would not have happened without me going through that process. . . But without any kind of, you know, research background, I'd have never done that." #6, M, 50-69yrs, participant & PPI.* |

*(Continued)*

**Table 3.** (Continued)

| Theme 4. Capturing research impacts | |
| --- | --- |
| 4.1 Agenda of people with the purse strings | *"Obviously it tends to focus on what's the saving for like The NHS."* #7, M, ≤49yrs, participant. |
| | *"Value for money aren't they. Is the money that we're going to put into this going to generate savings either in monetary terms or quality of life terms."* #21, M, 70-89yrs, PPI. |
| | *"The purse string holders, it is always about how you can provide what they want, as well as what you want for improving the service."* #1, M, 50-69yrs, participant. |
| 4.2 Real-world impacts | *"If it is a reduction in treatment burden on people that I guess can be estimated, whether it be hours per week per patient or something like total number of. . . and that has a benefit, a wider societal benefit doesn't it that can be translated. . . you know, what does that mean in terms of helping patients or helping the people that work for the NHS."* #7, M, ≤49yrs, participant. |
| | *"You've done all the bits that the men in the suits and the officials want, but a little side pod off that is the fact that I guess you'd sum it up as you'd improved the quality of life, but to me it runs a lot deeper than that."* #6, M, 50-69yrs, participant & PPI. |
| | *"Obviously you get it into the professional press, and it's read by other professionals, but it isn't taken up by the people, with that sort of intervening administrative level, between the researcher and the eventual [patient/service user]."* #22, F, 70-89yrs, PPI. |
| | *"I think the clinical findings is obviously the most important thing, but superimposed on that, it's always useful, dare I say, to have a picture of somebody who's participated. You know, if you imagine that little bit of research becoming on the BBC News website, you know, they'll look at that, or it goes in the Daily Mail or something, and it'll say, you know, "Jemima participated in this," and that brings it to, "Ah, that person is. . ." otherwise it's a bit dry, isn't it."* #2, M, 50-69yrs, participant. |

M male, F female, yrs years, PPI patient and public involvement

*"There has to be a need. It's going to have to serve some purpose. . . I'll tell you what I also totally dislike. . . you'll have hundreds of [research] students doing the same thing."* #21, male, 70–89 years, PPI.

To create meaningful research and add to the existing body of evidence, interviewees with PPI experience highlighted the importance of engaging patients and the public in the early stages of a research project. This was viewed as essential to establish appropriate research questions and design.

*"So that was very early stages, in terms of, "Are we approaching this correctly?" And the answer is no you weren't, and they agreed. But there are other cases where, yes, you're approaching it correctly, but the wording needs to be changed, it needs to be less scientific. You're needing to broaden the scope to involve more people, different age groups, different ethnicities, different sexes etc."* #21, male, 70–89 years, PPI.

However, one interviewee was concerned that involving patients or the public in research design might lead to too many competing opinions:

*"As soon as you start to involve and recruit the general public to help you design your research, then you are in for a bumpy ride. . . you are going to get all sorts of answers".* #5, male, 70–89 years, participant.

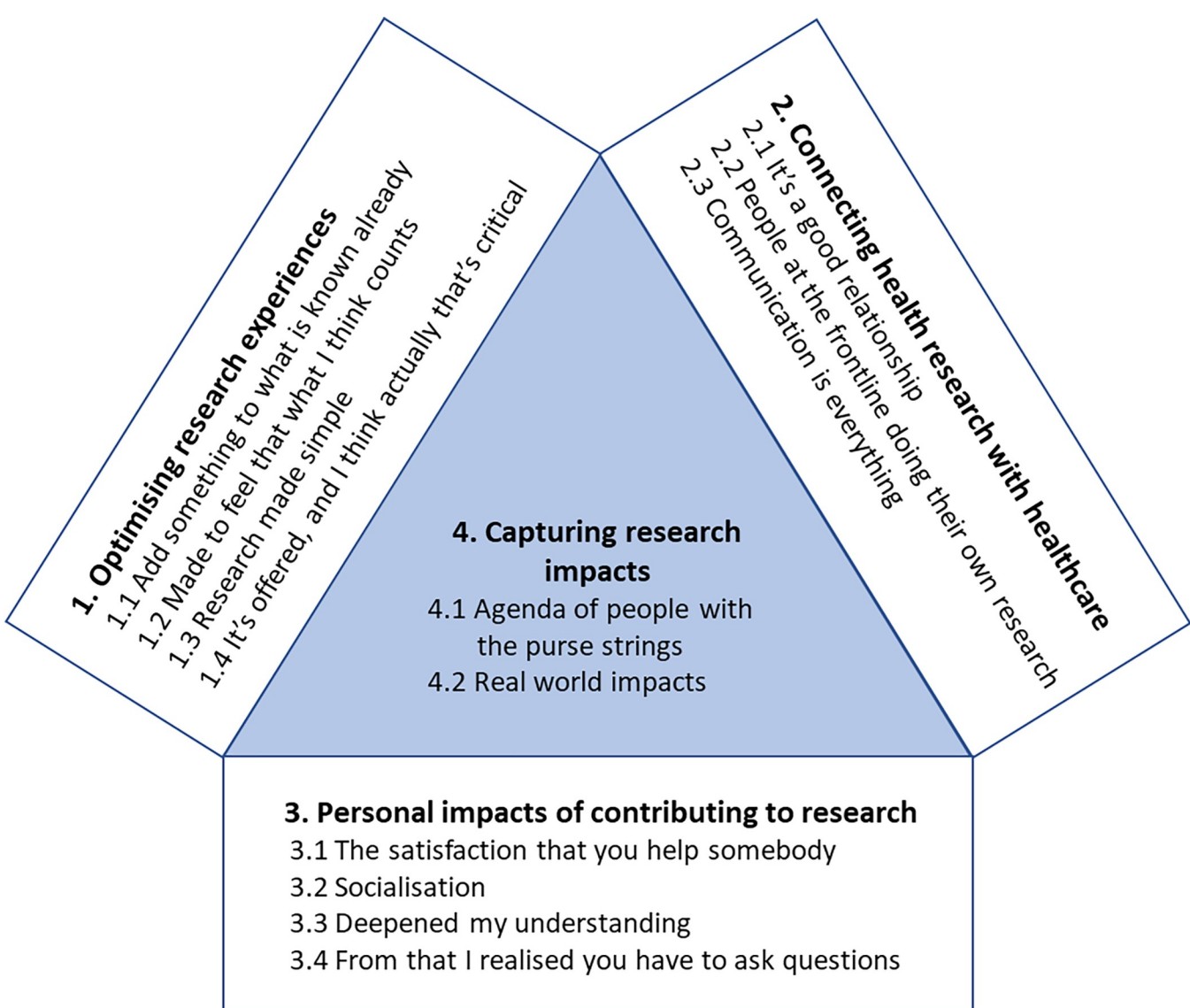

**Fig 1. Summary of the identified themes.** Themes 1 and 2 represent strategies for health researchers to optimise the experience for contributors and improve the connection between health research and health care. Theme 3 reports the perceived impacts of contributing to research for those who volunteer as participants or public advisors. These three themes converge on the final theme (theme 4) which describes the types of impact that were identified as important.

**1.2 "*Made to feel that whatever I think counts*" (#20).** Interviewees recalled positive interactions with researchers, highlighting that they felt valued for their contribution. They suggested that this should be a priority for all research. Feedback after research completion was viewed as a crucial component of feeling valued, with some interviewees expressing frustration when they did not find out the ultimate findings of a study in which they had been involved. This included any subsequent research and changes to practice.

*"The only way you can give a reward is to let them know what they're doing is so important or they are so important, it's keeping to the individual."* #19, female, 70–89 years, PPI.

*"And then from our point of view, to make us feel as if we're loved and appreciated, a catch-up letter every three months or six months is magic, but the important bit is when the research*

*has finished, a letter telling us what the outcome was, and then the researchers, where have they gone once the research has finished?" #13, male, 70–89, participant and PPI.*

**1.3 "Research made simple" (#20).** Interviewees commented on the flexibility of booking research appointments or PPI meetings, with options to arrange times to fit with other activities. This included offering in-person appointments at times when public transport was cheaper or free with travel passes. Additionally, several interviewees suggested that researchers should orientate participants to the expected activities and timings for any research participation. These were seen as simple steps that would facilitate research contributions and optimise participant experience.

*"Retired people, we could get there for nothing, effectively, as long as it was after 10 o'clock in the morning." #22, female, 70–89, PPI*

*"I didn't realise there wasn't a gap in the time to have a sandwich. I remember saying, 'It's lunchtime now and I'm really quite [hungry]. . . So sometimes just a little idea of how the day might progress. It is quite nice, you know, just to take you through the day, because some people would be really worried about that." #2, male, 50–69 years, participant.*

Despite widespread positive experiences, many interviewees expressed the need for improvements to the documentation they were required to read and sign as part of research participation. This included confessions from some that they did not read the consent forms, being put off by the format and wording. Many interviewees suggested that research documentation should be simplified to make it more user-friendly.

*"Some of them need to be better thought-out, I think some of them just go on and on and on! So, yes some of them you think are far longer than one would imagine they ought to be." #5, male, 70–89, participant.*

*"I always like it simpler. I think everyone thinks it's good to try and keep it simple if possible. Obviously, some of it is tends to be boiler plate legal, so I can kind of understand that they want to put that on, or they have to put that on. . . . It's very much like when you're installing some new software and you scroll right to the bottom and say yes and you've got no idea what you've read." #8, male, <49 years, participant.*

Finally, being able to contact the research team easily in the case of any questions or queries was highlighted by one interviewee as a simple strategy to improve the research experience.

*"If the only person who's left their contact details on the application are the people in charge, sometimes they're a bit hard to get hold of. . . So if there is a link between me, the member of the public, and the researcher, that also is good." #13, male, 70–89, participant and PPI.*

**1.4 "It is offered, and I think that's actually critical" (#21).** Most interviewees recalled that they had been offered expenses or reimbursement for their research contribution. Many did not accept the payment, reporting that it was not necessary on those occasions, and that they were happy to give up their time. However, there was awareness that not everyone has financial security, and that appropriate and timely reimbursement may facilitate broader research participation.

*"Lots of people said, 'No, I don't want that money, put it back into the study', others wanted it donated, most people didn't want it, I'd say. Reimbursement for actual costs, like train rides if you're having to attend something, and maybe providing lunch, that type of thing is great, but I think mostly people don't want, don't need paying per se to take part in things. #15, female, <49, participant and PPI.*

*"I do it because I feel I gain a lot of interest myself, so I don't really want to be paid. I think what, to me, is so right with that is that they have managed to recruit people who are ill and need money because of their ailments and inability to work and earn a good living, it is, in the worst of senses, pocket money, but in some of them, I think it's an added financial benefit to do it even if it's only for two or three hours a month." #16, male, >90, participant and PPI*

Payment was also seen as a method of demonstrating the value of research participants and PPI members.

*"'You are very important. Your data, your input is going to change the way that this project is developed and without your input the project won't go ahead so it's right that you are reimbursed for your time and what you put in', which was quite refreshing to me because from an NHS point of view, you're always treated as a patient. So, to have that recognition was good."* #6, male, 50–69 years, participant and PPI.

A small number of interviewees made a distinction between research within the National Health Service and research by organisations and companies who stood to profit financially. In the latter case, payment for participation appeared to be more acceptable.

*"The only time I've actually felt comfortable about taking money is where I've been doing things for a commercial company. . . if a commercial company is actually getting money from what we participants have given them, I think it's fair enough that they can give us a little something."* #22, female, 70–89 years, PPI.

## 2. Connecting health research with healthcare

Interviewees described positive interactions with their research teams, with many listing research-active clinicians by name and explaining how their research contribution led to the development of these relationships *(2.1 It's a good relationship)*. Research was viewed as important to the continued development of all aspects of healthcare and interviewees highlighted the value of clinicians being involved in research *(2.2 People at the frontline doing their own research)*. However, some interviewees identified that there was a lack of communication between research teams and other aspects of the health service *(2.3 Communication is everything)*.

**2.1 *"It's a good relationship"* (#11).**   All interviewees reported that their research contribution had been a positive experience. They recalled building relationships with the researchers based on mutual respect. These appeared to contribute to a sense of value for both the participation and the process of healthcare research, and contributed to relationships with broader clinical teams.

*"I think it has built it [the relationship with the clinical team] stronger because we've always had a connection to the [specific hospital allied health] department through the charity anyway and I think that through that, why she felt confident to approach us to actually have some advisors on the sort of study and I think it has definitely reinforced that. So, it's a good relationship now with them."* #10, male, <49 years, participant and PPI.

*"If I am contacted, I know it's through the [named hospital] in some way, so I know I can trust that it's going to be legitimate".* #5, male, 70–89 years, participant.

**2.2 *"People at the frontline doing their own research"* (#13).**   Interviewees advocated clinician involvement in health research. Frontline clinicians were identified as key individuals who were able to identify issues and research questions relevant to their area of practice. Clinicians were entrusted with improving things for patients and service users, and therefore able to make meaningful contributions to health research.

*"I think that does produce the best research. Because if you just had people that weren't actually on the frontline implementing this sort of stuff, seeing it day to day. . . they're not quite in touch with what they're researching, whereas the [frontline] people, that is actually their bread and butter."* #10, male, <49 years, participant.

Several interviewees noted that there was likely to be a tension between clinical and research duties, but emphasised that allocated research time was important for the progression of services. It was acknowledged that not all clinicians have an interest in research, but that it would be beneficial to support those who did.

*"I think the research is valuable and it should be timed in, so like put it into their schedule, 'that's research time, that's research time, that's clinical time', done like that."* #9, female, <49 years, participant.

*"I think not everybody does, not everybody's that personality. Some people are just happy doing their clinic work and don't want to get involved in the future, you know, the future learning and progressing. But I think those who are and are passionate and genuinely adding value, you know, it should be part of their role, I think. It's hard to say, but yeah, I think that's definitely value."* #1, male, 50–69 years, participant.

One interviewee highlighted that the administrative staff within clinical departments should have the opportunity to be involved in health research. It was acknowledged that these individuals also play an important role in health service delivery.

*"The administration, they are never included. . . There's aspects of what makes the machinery work in the NHS, there's many aspects, and you have people who are dying to say what's wrong because they know something needs putting right, but they're never involved in this new research that's being worked on. And that can be a stumbling block to the success of research. Their input is as valuable as the public and patients."* #19, female, 70–89, PPI.

**2.3 *"Communication is everything"* (#9).**   Despite widespread positive experiences, several interviewees identified a need to improve information sharing between research teams and participants or their clinicians. It was suggested that two-way communication was something that could be improved to optimise both participant experience and the utility of their research data as part of their broader healthcare interactions.

*"It was a really interesting study, which was multiform and multisource, but, things like the [physiological measures] didn't go back to the doctors, there wasn't the two-way flow, it was*

*just a single way, but I think it would have been useful in the future to be able to do that."* #15, female, ≤49 years, participant and PPI.

*"Communication is everything. Communicate. Yeah, just communicate with people, like with myself and also within your team and do it that way. So for the last research I did, I didn't know if I was still in it, 'am I still doing it?'. So communication broke down."* #9, female, ≤49yrs, participant.

### 3. Personal impacts of contributing to research

All interviewees reported personal benefits of being involved in health research. These were described in four broad sub-themes. The first centred on the personal satisfaction of contributing to research that would benefit others *(3.1 The satisfaction is that you help somebody)*. The second related to the social aspect of being involved and meeting others *(3.2 Socialisation)*, while the third related to increased knowledge and ongoing learning *(3.3 Deepened my understanding)*. The final sub-theme described the application of this learning and networking through increased confidence when interacting with health or other social services outside the research environment *(3.4 From that I realised you have to ask questions)*.

**3.1 *"The satisfaction is that you help somebody"* (#20).**  Most interviewees recalled that they took part in research to help future patients and/or to give something back to their hospital or clinical team. This was viewed as worthwhile and was associated with enjoyment and satisfaction.

*"Due to my injury and the care I've had through the NHS, I really want to give something back. So, I'm part of a charity organisation that helps people with similar injuries and brings to light this sort of thing. And I try and throw myself in as much as I can. So, anything crops up like that. . . I just grab it with both hands. It feels good and I really like the idea of being able to help people."* #10, male, ≤49 years, participant and PPI.

*"The word 'volunteering' makes it sound selfless, but it's about doing something that you get a buzz out of having done."* #12, male, 50–69 years, participant.

**3.2 *"Socialisation"* (#22).**  The social benefits of taking part in research were predominantly discussed by retired interviewees and focused on social interaction, well-being and developing new networks. However this was not unique to retirees and several younger interviewees also recalled how they had benefited from hearing the stories of others with the same health condition.

*"I always enjoy going up to the hospital, they are such lovely people up there and it is quite a nice outing for me!"* #4, male, 70–89 years, participant.

*"It was nice to meet other people with the same problem. They might come up with a question that you think, 'oh yeah, that's interesting, I wish I'd thought of asking that'. Or the answer might help you with a question that you may have had."* #3, female, 50–69 years, participant.

**3.3 *"Deepened my understanding"* (#1).**  Most interviewees recalled that an important impact of their research contribution had been a deeper understanding of their health condition or increased knowledge about health more broadly. This was achieved through discussion with the research team, additional clinical assessments and feedback of the results.

*"I find it invaluable to be part of it, to see what things are progressing."* #3, female, 50–69 years, participant.

*"So, yeah, part of my reason for doing it obviously beyond helping them out and sort of helping the research was actually the fact that I hadn't really had any kind of proper tests done for a while"*. #7, male ≤49 years, participant.

Retirees also mentioned that contributing to research gave them the opportunity for life-long learning and mental stimulation:

*"If you're retired, your brain suddenly gets switched off. You've had an interesting job, and suddenly, 'oh, it's gone', and you just want some mental stimulus, and this is great."* #22, female, 70–89 years, PPI.

*"You should never stop learning, and I think I now understand, or at least comprehend, rather than understand, more of how the body works, than I ever did in my ordinary life."* #16, male, ≥90 years, participant and PPI.

**3.4 *"From that I realised you have to ask questions"* (#18).** In addition to increased knowledge and understanding about health, interviewees also recalled how they were able to apply the learning and skills that they had acquired to other settings. This included feeling more confident discussing treatment options with their clinicians or advocating for family members in education or healthcare environments.

*"When I go to the school, I will be a lot more into 'well what's happening here now, why are these students doing this, what is the background to it'. . . And, who've they involved and have they asked the parent's opinions and things like that–you just become more aware and I sup-pose because you are asking those questions."* #18, female, 50–69 years, PPI.

*"Before I took part in being a volunteer. . . I wouldn't challenge consultants or doctors whereas now I can do it in a relaxed way. Before I would have had to do it in so like a bit of frustration, whereas now I can ask before I get to that point of, you know, frustration so I've learnt that."* #19, female, 70–89 years, PPI.

## 4. Capturing research impacts

Interviewees highlighted the complexities of identifying and recording impact and acknowl-edged that different stakeholders would have different interests in the processes and outcomes of health research. Interviewees identified a distinction between the types of impact that might be important to commissioners *(3.1 Agenda of people with the purse strings)* and *Real-world impacts (3.2)* that included patient benefit, progression of knowledge and wider societal gains.

**4.1 *"Agenda of people with the purse strings"* (#1).** Several interviewees suggested that there may be specific research impacts that were important to those in charge of health ser-vices. These individuals were referred to as people holding the purse strings (#1) and men in suits (#6). It was suggested that their primary focus was on value for money and cost savings, and that reporting of research impacts needed to be targeted to these needs.

*"I mean commissioners love objective data! Less visits, less time spent with other services, that kind of data is high on the agenda of people concerned with the purse strings. . . even turn*

*subjective opinions into objective data, to keep the commissioning people happy."* #1, male, 50–69 years, participant.

*"I guess that comes down to convincing your managers of the value of the research. If they agree that it is something which is vital to know? I don't know, that is what you are up against, isn't it?"* #5, male, 70–89 years, participant.

**4.2 "*Real-world impacts*" (#12).** Potential impacts of health research centred on enabling better clinical outcomes for patients or service users as part of the research, and wider implementation of the research findings. Several interviewees also stressed that negative findings were just as important as positive results, specifically through eliminating the use of unnecessary or ineffective treatments or *debunking theories* (#15).

*"It's patient-reported experience and outcome measures, you know, it's the proof of the pudding is when the patient gets to eat it, in effect, and how well it tastes for them."* #11, female, 50–69 years, participant and PPI.

*"Or proven to be of no use, in which case that is a useful result."* #2, male, 50–69 years, participant.

The need to appropriately communicate the research findings was also emphasised. Interviewees indicated that the results should be shared with the public, as well as other healthcare professionals, and in an engaging format. Widespread knowledge of the research findings was perceived as an important impact and that effort was needed to support dissemination.

*"And what's needed is to build up over the years a series of links so that other important organisations, significant organisations, are following what you say so that when you do make a tweet with a link to some new research finding, that can get picked up and maybe before you know where you are you're being interviewed on radio, and it's being picked up by the various media outlets if you've managed to turn it into 140 interesting characters."* #12, male, 50–69 years, participant.

*"The public can't necessarily cross the line to come into the science community because when you start going all scientific on me, I'm like, 'I haven't got a clue what you're talking about'. However, you have the ability to cross the line to come to me to say, 'right, we've done all this research. You might like it. It meant this'."* #6, male, 50–69 years, participant and PPI.

A few interviewees mentioned traditional academic impact metrics, such as publications, presentations, and completion of academic qualifications. However, the implementation and progress of the research findings were perceived as more important.

*"You don't want–all the boxes to be ticked, 'ra, ra, ra', someone's gone on stage and done their nice presentation, but now it's forgotten and they've moved on and they're getting on with their job and not doing anything to do with the research, and nobody else is either."* #15, female, ≤49 years, participant and PPI.

Interviewees also demonstrated the impacts that they and other patients or members of the public had made on the research they were involved in. This included refining research questions and strategies to improve research recruitment. One interviewee suggested keeping a record of these contributions to illustrate the impact.

*"I think it is probably easier having taken notes during a meeting, to immediately enter it into a log of so-and-so suggested this and we have incorporated it, or so-and-so has suggested that and further down the line you follow up and say yes this made this difference."* #23, female, 50–69 years, participant and PPI.

## Discussion

This qualitative interview study explored the experiences of research participants and PPI group members from a range of health-related studies led by different research-active NMAHPP clinicians. The aims were to understand more about the experiences of contributing to health research and to uncover the types of impacts that are important for these individuals. Interview content was summarised in four main themes: 21) optimising research experiences, 2) connecting health research with healthcare, 3) personal impacts of contributing to research, and 4) capturing research impacts.

In the first theme, interviewees provided several suggestions and strategies to optimise the experience of participants and PPI members, and ensure that the research generated societal benefit. This included ensuring that the research focus was meaningful to those involved and that everyone's contribution was valued. The importance of feeling valued was a salient theme in a recent study of PPI reporting [44]; interviewees in the current study emphasised that this was also key for research participants. All interviewees identified the importance of feedback to confirm that their contribution was valued. This included regular updates if the study continued over an extended period, and a summary of the findings. Regular updates were also endorsed by cancer trial participants in a survey of their feedback requirements, although research teams preferred to communicate feedback only at the end of the study [45]. Possible reasons for the reluctance of research teams to provide regular updates include the time required to prepare the material and concerns about sharing preliminary (and therefore unpublished) findings. However, we found that participants and PPI members appear to appreciate the courtesy of regular updates and suggest that this could be included as part of research dissemination plans.

As well as feedback at the end of the study, interviewees were interested in what happened next, for example whether their contribution led to changes in practice or whether the research had progressed. Feedback about these later impacts might be more challenging for researchers because the main modes of funding NMAHPP research are short-term grants or fellowships [28]. Once the study is complete, the researcher may no longer have access to participants' contact details or be permitted to contact them. Furthermore, they may not have allocated time for research activities. We encountered both issues when exploring recruitment options for the current study [46]. We suggest that consent forms should include a question asking about consent for contact about future research and for updates on the research after the study is completed, and that researchers should factor in the need to communicate later impacts.

An aspect of the second theme *(2.3 Communication is everything)* also links with this discussion around information provision and communication. Interviewees suggested that data from their research assessments might be useful for their clinicians in primary care or other clinical settings, but that this was not accessible. Again, this would require alternative consent procedures and strategies to support safe information transfer.

Interviewees advocated for simpler research documents. Several interviewees recalled that they did not read research material, including consent forms, because the unwieldy length and format were off-putting. This raises questions about truly informed consent, and also highlights that we, as researchers, need to do better in co-designing user-friendly research

materials. A recent consensus exercise involving researchers, ethics committee members and PPI members generated evidence- and experience-informed guidance for designing accessible participant information sheets and consent forms [47]. This included layout, formatting, language, content and readability checking. This guidance has yet to be empirically assessed, but may be a good starting point for preventing some of the issues raised by interviewees in the current study.

The final component of optimising the research experience focused on money. Interviewees reported that they were not driven by financial incentives, which mirrors findings from a recent systematic review of research participation [48]. For those with more extensive PPI experience, payment was an important demonstration of the value that people with lived experience bring to research, particularly for studies where there would ultimately be financial gain for the research team. This study was conducted within the context of the UK National Health Service, which is free at the point of use, and may therefore not reflect the views of participants and PPI members in countries where healthcare is not free to access.

The second theme explored connecting health research with healthcare. There was a clearly perceived beneficial impact of frontline clinicians being involved in research. This included a role in the timely identification of research questions and the subsequent implementation of research findings into practice. Several studies have reported an association between research activity and better patient outcomes [7–10]. Interviewees in the current study were specifically discussing NMAHPP clinicians and the findings illustrate experiential awareness that clinical research is not just the remit of clinical doctors. This further supports the international drive to develop NMAHPP clinical academic roles [11–18]. However, despite increasing research fellowship opportunities for NMAHPPs [49], there are persisting barriers to developing permanent clinical academic careers [22]. Interviewees' suggestions of allocated research time would be a welcome addition for many NMAHPP clinicians.

All interviewees reported personal benefits of being involved in health research (theme 3). We appreciate that our sample will be biased towards those with positive research experiences, as we were asking people to take part in additional research, however we gained some important insights into the perceived impacts of contributing to research. A desire to help others has been commonly identified as a key determinant of the decision to participate in research [48]. Interviewees in the current study also reflected on the personal benefits of learning more about their condition or health more broadly, and the social aspects of developing new networks. Interestingly, several interviewees recalled how their research experience led to increased confidence when interacting with health or social care professionals in other settings. Knowledge of research processes and experience of interacting with research teams appeared to support these individuals to have more autonomy in their own care or that of their family members. In addition to increased knowledge, this also appeared to be the result of empowerment through contributing to research with a realisation that their opinions hold value. This aligns with the concept of shared decision-making within healthcare [50, 51].

The final theme explored how the impacts of contributing to research might be captured. Interviewees stressed that different stakeholders are likely to have different needs and therefore multiple metrics might be required [52]. The personal impacts that were identified when discussing research experiences did contribute to recommendations for impact capture, although interviewees largely focused on impacts at a broader organisational or societal level. There was a particular emphasis on the *"men in suits"* or those *"holding the purse strings"*, and it was recognised that commissioners and managers were likely to have a focus on cost-effectiveness or cost-savings. However, interviewees' suggestions of real-world impacts centred on quality of life, clinical outcomes or measures of health status, and changes to healthcare delivery or

practice. These findings echo those of recent studies looking at participants' views of taking part in research and of participant retention [48, 53].

The next stage of our programme of work involves developing a framework to support a consistent method of capturing the important impacts of NMAHPP research. The current study has informed this process by highlighting impacts that are important for participants and PPI members. In particular, the provision of progress updates and feedback during the study, on completion, and following the study. Access to this information has been highlighted as important for participants and a method of enhancing the experience of research. This gives researchers the opportunity to reflect on their achievements and outcomes at different time-points to ensure that they are generating the types of real-world impacts that were suggested by interviewees.

## Limitations

The current study included both PPI members and research participants, however the majority of participants had been involved in both roles. It is possible that a study population comprising solely of research participants may have different experiences and different views on research impacts. However, the aim of the study was not to differentiate between the views of PPI members and research participants, rather to look broadly at the types of impact that were considered valuable by these individuals.

Importantly, there was limited ethnic diversity among our study population; interviewees were predominantly White or White British. This is a recurring issue with healthcare research [54], and we were potentially limited through our recruitment of those already involved with health research. Future research should specifically seek the views of individuals from minority populations to ensure that their opinions are incorporated into the ongoing discussion around research experiences and impact.

The number of interviews that were conducted (n = 21) was guided by a blended definition of data saturation; no new concepts identified during the last 2–3 interviews and no new codes identified during analysis. The sample size calculation described in the protocol and application for ethics approval was based on a feasible range with our available resources (n = 15–25). There are ongoing discussions within the field of qualitative healthcare research regarding the assessment of sample size, including whether data saturation is an appropriate concept and how this might relate to the credibility of the findings [40, 55]. We are confident that our sample size and approach maximised the trustworthiness and authenticity of our findings [56], while acknowledging the limited ethnic diversity, as outlined above. As discussed previously, recruitment for the current study was limited by researchers not having permission or access to contact previous participants [46]. This led to the use of wider social media recruitment strategies, which limited targeted recruitment based on demographic or other factors.

We aimed to specifically recruit individuals who had been involved in research led by NMAHPP clinicians; unfortunately, and despite our best efforts, we were unable to recruit participants involved in nurse or midwife-led research and these clinical groups are unrepresented. In addition, many interviewees had been involved in multiple studies, some of which had been led by clinical doctors or university-based academics. For this reason, we suspect that our findings will apply more broadly and may not be limited to NMAHPP clinical academic research settings. The issues and suggestions that were raised by interviewees were not unique to research led by NMAHPPs or research within these clinical areas. This is an unplanned strength of the current study, and as such, we hope that the findings will be useful for researchers beyond NMAHPP research settings.

## Conclusions

In conclusion, individuals who were involved in NMAHPP health research, either as participants or PPI members, recalled positive experiences of their contributions to research and reported good relationships with their research teams. They felt that their contributions were valued. Suggested strategies to optimise the research experience focused on simplifying documentation, clear signposting of the research activities involved, and feedback on the research findings, including how this was applied in clinical settings or used to develop new research. Routine sharing of relevant research data with clinicians was also recommended to aid the translation of research finding into practice. Personal impacts of contributing to research included deeper understanding of their health condition or health more broadly, and increased confidence interacting with healthcare and other professionals. Locally, these findings will be used to inform development of a framework to capture the impact of NMAHPP research. We also hope that the findings will be applied more broadly, with the aim of improving the research experience for participants and PPI groups, and emphasising the types of research impact that are most meaningful to these contributors.

## Supporting information

**S1 File. Standards for Reporting Qualitative Research (SRQR).**
(PDF)

## Acknowledgments

The authors would like to thank all interviewees for giving up their time to participate in this study and their willingness to share their experiences. We would also like to thank our patient advisory group for their support in developing the study: Jill Lloyd and Flavio Affinito. Study recruitment was supported by the following clinicians and healthcare research teams: Gemma Clunie, Gemma Stanford and Adine Adonis, Imperial College Healthcare NHS Trust; Caroline Miller, University Hospitals Birmingham NHS Foundation Trust; Yeliz Prior, University of Salford; Anthony Gilbert, Royal National Orthopaedic Hospital NHS Trust; Jodie Bloska, Cambridge Institute for Music Therapy Research; Katherine Grady, Research for the Future; Shawn Walker, Chelsea and Westminster Hospital NHS Foundation Trust; and Halle Johnson, Imperial College London on behalf of VOICE. Finally, we would like to thank the peer reviewers for their helpful feedback and recommendations.

## Author Contributions

**Conceptualization:** Lisa Newington, Caroline M. Alexander, Mary Wells.

**Data curation:** Lisa Newington.

**Formal analysis:** Lisa Newington, Pippa Kirby, Rhia K. Saggu.

**Funding acquisition:** Lisa Newington, Caroline M. Alexander, Mary Wells.

**Investigation:** Lisa Newington.

**Methodology:** Lisa Newington.

**Project administration:** Lisa Newington.

**Supervision:** Caroline M. Alexander, Mary Wells.

**Writing – original draft:** Lisa Newington.

**Writing – review & editing:** Caroline M. Alexander, Pippa Kirby, Rhia K. Saggu, Mary Wells.

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
