## [Decision Letter · Decision Letter 0]

12 Aug 2022

PONE-D-22-17688Reflections on involvement in health research: a qualitative interview study with research participants and patient advisors.PLOS ONE

Dear Dr. Newington,

Thank you for submitting your manuscript to PLOS ONE. After careful consideration, we feel that it has merit but does not fully meet PLOS ONE’s publication criteria as it currently stands. Therefore, we invite you to submit a revised version of the manuscript that addresses the points raised during the review process. The manuscript has been evaluated by three reviewers and their comments are outlined below. 

We look forward to receiving your revised manuscript.

Kind regards,

Steph Scott

Academic Editor

PLOS ONE

Journal Requirements:

2. You indicated that you had ethical approval for your study. Please clarify whether minors were involved in this study. If so, in your Methods section, please ensure you have also stated whether you obtained consent from parents or guardians of the minors included in the study or whether the research ethics committee or IRB specifically waived the need for their consent.

 "LN was funded by an NIHR Imperial Biomedical Research Centre (BRC) and Imperial Health Charity Postdoctoral Research Fellowship (RFPO2122_3). 

RKS was funded by an NIHR Imperial BRC Pan-London Research Fellowship (ICHT-39). 

PK was supported by Imperial Health Charity, MW and CMA were supported by NIHR Imperial BRC. "

5. Please expand the acronym “BRC” (as indicated in your financial disclosure) so that it states the name of your funders in full.

Reviewers' comments:

Reviewer's Responses to Questions

**Comments to the Author**

1. Is the manuscript technically sound, and do the data support the conclusions?

Reviewer #1: Yes

Reviewer #2: Partly

Reviewer #3: Yes

2. Has the statistical analysis been performed appropriately and rigorously? 

Reviewer #1: N/A

Reviewer #2: N/A

Reviewer #3: N/A

3. Have the authors made all data underlying the findings in their manuscript fully available?

Reviewer #1: No

Reviewer #2: Yes

Reviewer #3: Yes

4. Is the manuscript presented in an intelligible fashion and written in standard English?

Reviewer #1: Yes

Reviewer #2: Yes

Reviewer #3: Yes

5. Review Comments to the Author

Reviewer #1: This is a valuable, interesting and well-written piece of research which will be of interest to a broad range of readers. I enjoyed reviewing this manuscript and believe, with some minor adjustments, it would meet the criteria for publication. As such, I have included a few suggestions below to strengthen this manuscript.

Overall:

Some of the language in the paper should be clarified – ‘involvement in research’ or ‘research involvement’ means something specific in the context of health research (i.e. research being carried out ‘with’ or ‘by’ members of the public). You use this term throughout the paper to refer to both participation and actual involvement. When discussing both concepts together you should use another term. For example ‘contribution to research’ (i.e. ‘..interviewees reflections on their contribution to research’).

Introduction:

Later in the paper you mention that the aim of the research was not to differentiate between the views of PPI members and research participants – while I understand your reasons for including both of these groups and collating their views, I do think it is important to be very clear about how these groups differ in the earlier stages of the manuscript. The NIHR provide concise, helpful definitions of the differences between research participation and research involvement and I suggest these should be explicitly referred to in the introduction section (see https://www.nihr.ac.uk/documents/briefing-notes-for-researchers-public-involvement-in-nhs-health-and-social-care-research/27371).

Methods:

In the abstract, you mention that you used Thematic Analysis (TA) but have not cited Braun and Clarke within the main paper and only given brief details of the steps that were taken. Please add more clarification within the methods section, around the method and process of data analysis. It may also be worth keeping in mind that data saturation is a contested concept within certain types of TA (see Braun & Clarke, 2021).

It would also be of benefit to add some more detail detailing the PPI contribution in relation to this project. For example, within the ‘research team and reflexivity’ section you state a public advisor contributed to the analysis, yet at line 123-126 there is mention only of the four authors. Can you provide further detail regarding how your public advisor was involved in this process?

Results:

Great to see lots of supporting quotations throughout the paper.

The sub-theme title for 2.3: ‘There wasn’t the two-way flow’ could more clearly convey the content of the theme (e.g. ‘Improving communication/information sharing between teams’ or similar).

Discussion:

This study appears to have two linked but distinct aims 1) to identify strategies to optimise research experience for members of the public and 2) to understand what research impacts are important to the public. In places it felt like these two aims were conflated somewhat e.g. line 520-524 – Whilst I agree that providing progress updates and feedback to public advisors/participants is crucial, I don’t think this can be described as a ‘research impact’ in itself – it is a strategy to improve research experience. Please more clearly summarise your findings in relation to these two separate aims.

Research impact (changes that come about from the research study itself e.g. clinical outcomes, changes to practice etc.) and impact of public involvement (changes that come about from the inclusion of the public in research e.g. increased confidence, more relevant and accessible research outcomes) are again slightly different things that often seemed to be slightly confused/conflated when reading the manuscript. It may help to briefly delineate these different ‘types’ of impact perhaps within the intro and/or discussion sections.

Line 545-546 – you frame this as a limitation but I would not particularly consider it as such? I think it is a good point to acknowledge, but readers of this study will come from a range of backgrounds and/or interested in public views of impact more generally. For these reasons I think these findings have implications beyond NMAHPP research settings so can also be considered a strength of the study.

Linked to my point above – I feel it would strengthen the discussion if you were to highlight how these findings could be useful/informative in the wider context (i.e. beyond just contributing to your framework). Public views around 1) improving the research experience and 2) important research impacts are valuable for health research more broadly!

Within the conclusion you state ‘Routine sharing of relevant research data with the participants' clinicians was also recommended’ (line 553-554). I did not see adequate evidence of this within participants quotes and it isn’t clear whether you mean sharing individual level data (which may raise ethical concerns) or broader research findings. I feel it would be more accurate/relevant to either refine your wording or emphasise the value of information sharing in the broader context (i.e. to further facilitate translation of research findings into practice).

Ethical consideration:

It was good to see public involvement was included with the study design, materials development and analysis but I was slightly disappointed that public members were not named as part of the authorship team (although I can see they were mentioned in acknowledgements). If public members were involved in drafting/approving the manuscript, please consider whether they would like to be named as co-authors in line with IMCJE guidelines. If not, this would certainly be valuable to consider as you move forwards with your ongoing programme of work.

Reviewer #2: Thank you for allowing me to review this interesting, insightful and well written manuscript. As a pharmacist researcher I found the narrative valuable for considering patient participant and PPIE needs/wants/requirements. The context for the study is adequately described with reference to appropriate literature from the NIHR. The findings are comprehensively reported, illustrated by a variety of quotes from across the sample. I found it particularly insightful to use short quotes as the 'names' for the sub-themes - this is something I have not seen before however ensures that the analysis is underpinned by the patient experience and true to the data. The discussion covers thoughtful topics of importance to NMAHPP researchers, such as the challenges we face with grants & resources and also provides some recommendations of how we can improve recruitment and retention of participants/ PPIE members, as well as make the experience of taking part in a study a more positive/ beneficial one.

I only have a few minor comments/ questions for the authors as highlighted below:

1. Please check author's affiliations. I could not see who was affiliated with '3. Nursing Directorate'.

2. Introduction, Pg. 3, line 56, 'demonstrate and evaluate the impact of this research involvement' - I was unsure what exactly the author's were referring to. Whose involvement? I am assuming PPIE/ patients but there has been no mention of them by this stage in the manuscript.

3. Introduction - It would be of value to describe why the focus on NMAHPP-led research, as oppose to all research (especially when research conducted by dentists & clinical doctors was excluded). What is it about NMAHPP research that is different, which has led to this piece of work?

4. Introduction, Pg. 3, line 63 should 'researcher' be plural?

5. Introduction, Pg. 4, lines 72-76 - I found this a rather large sentence. Suggest breaking up to improve readability if possible. Maybe at 'however'?

6. Research team and reflexivity - Pg. 5. I found this section to be lacking in any reflexivity. The backgrounds of the team were presented and their experience alluded to, however the section is missing what the impacts of this are on the study's conduct/ how this has influenced you. I also wondered if having completed your other work with clinicians might have given you some a priori insight and whether this influenced your data collection/ analysis? Did any personal/ professional beliefs incidentally affect this research? (S6 of SRQR checklist - need to incorporate some narrative around how these factors may have influenced the research/ researcher)

7. Materials and Methods - was any theoretical lens/ perspective used to inform your study? (S5 of SRQR checklist; cannot identify within narrative)

8. Study design, Pg. 6, line 114 - should 'or their involvement' be 'of their involvement'?

9. Study design, Pg. 6 - I would have valued some content around the methods of data analysis. From the abstract and limited detail it sounds as though code book thematic analysis was used, however a description of how the codes were managed into the final thematic map would be useful. It would also be useful to know if PPI data has handled separately to patient participant data. Within the findings differing perspectives were offered between these groups and I wondered how you managed this in the analysis.

10. Discussion, Pg. 24, lines 463, 464. This sentence around funding doesn't quite make sense/ is hard to read. I understand what you are trying to say, but think the structure of the sentence isn't quite right. I also suggest that you make it more explicit how the challenges of funding/ backfill mean that it is difficult to feedback about changes in practice.

11. Discussion, Pg. 25, line 486, should be 'within the context of...' not 'with'

12. Figure 1. I can see that the blue colour of theme 4 represents (I think) the convergence of the 3 themes, however I am unsure exactly what the arrows represent. Are they crucial to the interaction of the themes in the analysis? If so, I would suggest this is made clearer.

Overall, I believe that this is a valuable piece of work for the NMAHPP community and has useful recommendations for all researchers working with patients/ PPIE. It is very well written and I would recommend it for publication with minor corrections. Thank you once again for the opportunity to review such important work.

Reviewer #3: This paper describes the findings of a qualitative research project investigating the experiences of patients and citizens who contributed to health research done by non-medical/dental health professionals (NMAHPPs). Its findings will inform a framework for evaluating the impact of NMAHPP research. The paper gives a comprehensive account of the study's design and results, supplying additional data to supplement the extracts that illustrate the findings. It is methodologically sound and will be an important addition to the literature on the value of involving people in health research. What is particularly noteworthy about this publication is its focus on NMAHPP health research specifically, an area that the authors acknowledge has yet to be as fully embraced by the relevant professionals as that in medicine or dentistry. Hopefully this project and its outputs will give future NMAHPP researchers confidence and a sound footing from which to plan their research with impact in mind. I look forward to seeing the forthcoming Framework.

6. PLOS authors have the option to publish the peer review history of their article (what does this mean?). If published, this will include your full peer review and any attached files.

Reviewer #1: No

Reviewer #2: **Yes: **Dr Justine Emily Tomlinson

Reviewer #3: **Yes: **Teresa M. D. Finlay

---

## [Author Response · Author response to Decision Letter 0]

5 Sep 2022

Please find our point by point response attached in the file ResponseToReviewers. 

Thank you for the helpful comments and recommendations. 

Very best wishes, 

Lisa

---

## [Decision Letter · Decision Letter 1]

13 Oct 2022

PONE-D-22-17688R1Reflections on contributing to health research: a qualitative interview study with research participants and patient advisorsPLOS ONE

Dear Dr. Newington,

Thank you for submitting your manuscript to PLOS ONE. After careful consideration, we feel that it has merit but does not fully meet PLOS ONE’s publication criteria as it currently stands. Therefore, we invite you to submit a revised version of the manuscript that addresses the points raised during the review process. Please see comments and feedback from all three reviewers below - please note, one reviewer still has a few small outstanding points of clarification for you to address.

We look forward to receiving your revised manuscript.

Kind regards,

Steph Scott

Academic Editor

PLOS ONE

Journal Requirements:

Reviewers' comments:

Reviewer's Responses to Questions

**Comments to the Author**

1. If the authors have adequately addressed your comments raised in a previous round of review and you feel that this manuscript is now acceptable for publication, you may indicate that here to bypass the “Comments to the Author” section, enter your conflict of interest statement in the “Confidential to Editor” section, and submit your "Accept" recommendation.

Reviewer #1: All comments have been addressed

Reviewer #2: (No Response)

Reviewer #3: All comments have been addressed

2. Is the manuscript technically sound, and do the data support the conclusions?

Reviewer #1: Yes

Reviewer #2: Yes

Reviewer #3: Yes

3. Has the statistical analysis been performed appropriately and rigorously? 

Reviewer #1: N/A

Reviewer #2: N/A

Reviewer #3: N/A

4. Have the authors made all data underlying the findings in their manuscript fully available?

Reviewer #1: Yes

Reviewer #2: Yes

Reviewer #3: Yes

5. Is the manuscript presented in an intelligible fashion and written in standard English?

Reviewer #1: Yes

Reviewer #2: Yes

Reviewer #3: Yes

6. Review Comments to the Author

Reviewer #1: The authors have thoroughly addressed all comments and I am more than happy to recommend this article be accepted for publication.

Reviewer #2: Thank you for the opportunity to review this manuscript again. The authors have mostly answered my questions and addressed my feedback. I have 3 further and final comments.

1. You now describe your theoretical underpinning as 'realist'. Please could you briefly describe how this perspective has shaped your work?

2. What particular factors guided your sample size? The manuscript mentions that sample size was determined by what was feasible. Was there anything else? Does your sample size limit or affect the quality or authenticity (https://doi.org/10.1016/j.sapharm.2020.02.005) of your work in any way? A comment in the strengths/ limitations section would be advisable.

3. You have mentioned that the analysis method was Thematic Analysis. Please be aware that current thinking is that there are 3 principle approaches to TA - Code reliability, code book and reflexive (see https://doi.org/10.1007/s11135-021-01182-y and https://doi.org/10.1080/2159676X.2019.1628806). The way it is currently described it sounds like code reliability - please could you accurately describe the approach to TA. This approach may then have implications for sample size (see point 2) and theoretical perspective (see point 1).

I hope this review is helpful.

Reviewer #3: (No Response)

7. PLOS authors have the option to publish the peer review history of their article (what does this mean?). If published, this will include your full peer review and any attached files.

Reviewer #1: **Yes: **Dr Angela Wearn

Reviewer #2: **Yes: **Dr Justine Tomlinson

Reviewer #3: **Yes: **Teresa M. D Finlay

---

## [Author Response · Author response to Decision Letter 1]

25 Oct 2022

Please see the attached document 'Response to reviewers'. This provides a point by point response and highlights the amended and additional text in the manuscript. 

Very many thanks,

---

## [Decision Letter · Decision Letter 2]

6 Dec 2022

Reflections on contributing to health research: a qualitative interview study with research participants and patient advisors

PONE-D-22-17688R2

Dear Dr. Newington,

We’re pleased to inform you that your manuscript has been judged scientifically suitable for publication and will be formally accepted for publication once it meets all outstanding technical requirements.

Kind regards,

Steph Scott

Academic Editor

PLOS ONE

Additional Editor Comments (optional):

Reviewers' comments:

Reviewer's Responses to Questions

**Comments to the Author**

1. If the authors have adequately addressed your comments raised in a previous round of review and you feel that this manuscript is now acceptable for publication, you may indicate that here to bypass the “Comments to the Author” section, enter your conflict of interest statement in the “Confidential to Editor” section, and submit your "Accept" recommendation.

Reviewer #1: All comments have been addressed

Reviewer #2: All comments have been addressed

2. Is the manuscript technically sound, and do the data support the conclusions?

Reviewer #1: Yes

Reviewer #2: Yes

3. Has the statistical analysis been performed appropriately and rigorously? 

Reviewer #1: N/A

Reviewer #2: N/A

4. Have the authors made all data underlying the findings in their manuscript fully available?

Reviewer #1: Yes

Reviewer #2: Yes

5. Is the manuscript presented in an intelligible fashion and written in standard English?

Reviewer #1: Yes

Reviewer #2: Yes

6. Review Comments to the Author

Reviewer #1: The authors have addressed all comments well, I look forward to seeing the article in publication.

Reviewer #2: Dear authors,

Thank you for addressing all of my comments and questions. The amendments you have made will help the reader to better understand your methods and rationale for certain decisions, which in turn will help others in their research. This is a really important manuscript and I wish you luck in whatever direction your project now takes you!

Best wishes,

Justine Tomlinson

7. PLOS authors have the option to publish the peer review history of their article (what does this mean?). If published, this will include your full peer review and any attached files.

Reviewer #1: **Yes: **Dr Angela Wearn

Reviewer #2: **Yes: **Dr Justine Tomlinson

---

## [Editor Report · Acceptance letter]

9 Dec 2022

PONE-D-22-17688R2 

Reflections on contributing to health research: a qualitative interview study with research participants and patient advisors 

Dear Dr. Newington:

I'm pleased to inform you that your manuscript has been deemed suitable for publication in PLOS ONE. Congratulations! Your manuscript is now with our production department. 

Kind regards, 

on behalf of

Dr. Steph Scott 

Academic Editor

PLOS ONE